# Candidate Biological Markers for Social Anxiety Disorder: A Systematic Review

**DOI:** 10.3390/ijms24010835

**Published:** 2023-01-03

**Authors:** Alice Caldiroli, Enrico Capuzzi, Letizia M. Affaticati, Teresa Surace, Carla L. Di Forti, Antonios Dakanalis, Massimo Clerici, Massimiliano Buoli

**Affiliations:** 1Department of Mental Health and Addiction, Fondazione IRCCS San Gerardo dei Tintori, Via G.B. Pergolesi 33, 20900 Monza, Italy; e.capuzzi1@campus.unimib.it (E.C.); teresa.surace@asst-monza.it (T.S.); massimo.clerici@unimib.it (M.C.); 2Department of Medicine and Surgery, University of Milano Bicocca, Via Cadore 38, 20900 Monza, Italy; letizia.affaticati@gmail.com (L.M.A.); diforticarla@gmail.com (C.L.D.F.); antonios.dakanalis@unimib.it (A.D.); 3Department of Pathophysiology and Transplantation, University of Milan, Via Festa del Perdono 7, 20122 Milan, Italy; massimiliano.buoli@unimi.it; 4Department of Neurosciences and Mental Health, Fondazione IRCCS Ca’ Granda Ospedale Maggiore Policlinico, Via F. Sforza 35, 20122 Milan, Italy

**Keywords:** social anxiety disorder, social phobia, biological markers, neuroimaging, systematic review

## Abstract

Social anxiety disorder (SAD) is a common psychiatric condition associated with a high risk of psychiatric comorbidity and impaired social/occupational functioning when not promptly treated. The identification of biological markers may facilitate the diagnostic process, leading to an early and proper treatment. Our aim was to systematically review the available literature about potential biomarkers for SAD. A search in the main online repositories (PubMed, ISI Web of Knowledge, PsychInfo, etc.) was performed. Of the 662 records screened, 61 were included. Results concerning cortisol, neuropeptides and inflammatory/immunological/neurotrophic markers remain inconsistent. Preliminary evidence emerged about the role of chromosome 16 and the endomannosidase gene, as well as of epigenetic factors, in increasing vulnerability to SAD. Neuroimaging findings revealed an altered connectivity of different cerebral areas in SAD patients and amygdala activation under social threat. Some parameters such as salivary alpha amylase levels, changes in antioxidant defenses, increased gaze avoidance and QT dispersion seem to be associated with SAD and may represent promising biomarkers of this condition. However, the preliminary positive correlations have been poorly replicated. Further studies on larger samples and investigating the same biomarkers are needed to identify more specific biological markers for SAD.

## 1. Introduction

Social anxiety disorder (SAD) is a common psychiatric condition characterized by a persistent and excessive fear of social situations in which a person may be subject to scrutiny by others [1]. SAD is often associated with other mental disorders, particularly depression and substance-use disorders [2], as well as with poor psycho-social functioning—including increased school drop-out, impaired workplace productivity and lower socio-economic status [3,4].

The treatment of SAD is based primarily on psychotherapy, pharmacotherapy, or both [5]. Among psychotherapeutic interventions, cognitive behavioral therapy (CBT) is considered as a first-line treatment (National Institute for Health and Care Excellence (NICE) guidelines) [6], with response rates between 50% and 65% [7,8]. On the other hand, selective serotonin uptake inhibitors (SSRIs) are considered as first-line compounds [1], although venlafaxine has evidence of efficacy for this condition [9,10]. Finally, benzodiazepines may be useful as initial or adjunctive treatment for patients that require rapid symptom relief [1,11].

The symptoms that characterize this condition (e.g., shame) hamper the pro-active request of help by these patients [1]. In addition, the frequent psychiatric comorbidity [2] may further delay a proper management of SAD with the risk that clinicians focus their attention on secondary, despite relevant, problems such as alcoholism [1]. In this context, the identification of clear-cut biomarkers may help in the identification of subjects affected by SAD and facilitate the differential diagnosis with other affective disorders, thus favoring the prescription of an early and proper treatment [12]. Furthermore, biological factors may have a role in treatment response both for SAD [13] and generally for anxiety disorders [14].

Putative biomarkers were described in several psychiatric conditions, including psychotic spectrum, mood and anxiety disorders [15]. The alterations involve different biological systems and the most studied biomarkers so far regarding anxiety disorders concern genetics, epigenetics, the endocrine system, the immune system, abnormalities in brain structure and function (neuroimaging) and neuropsychological aspects [15,16].

Current literature shows that anxiety disorders are complex and polygenic, although few genetic variants associated with an increased risk for these conditions have been identified until now [17]. In particular, a possible association between serotonergic and dopaminergic polymorphisms and SAD has been suggested. Similarly, some epigenetic mechanisms including DNA methylation have been hypothesized to be mediators of environmental and genetic risk factors for anxiety disorders, especially with regard to the individuals’ ability to implement coping strategies to face stressful events [18]. As regards SAD, a role of variants of the oxytocin gene (*OT*) was specifically underlined, given its influence on social functioning and interpersonal behavior. In humans, OT plays an important role in recognition, trust, parent-infant bonding and romantic attachment [19]: aspects that can be compromised in patients affected by SAD [20]. Some authors investigated polymorphisms within the gene coding for the OT receptor (*OXTR*, located at 3 p25.3) in families at risk of SAD, finding an association with a specific polymorphism (rs2254298) and with lower salivary oxytocin levels [21]. In addition, some authors reported an altered *OXTR* methylation in subjects affected by SAD [22]. Moreover, functional neuroimaging studies correlated an increased sensitivity towards social cues with *OXTR* polymorphisms [23].

Regarding endocrine biomarkers, hypothalamus-pituitary-adrenal (HPA) axis plays an important role in social interactions, regulating behavioral responses to stressors [24]. Some studies, though not all, have so far suggested that patients with SAD may differ significantly from healthy controls (HC) concerning baseline cortisol levels, and/or may demonstrate exaggerated cortisol stress-responsiveness possibly linked to increased social avoidances [25]. Furthermore, the literature has also focused on inflammatory mediators including C-Reactive Protein (CRP) and different cytokines, as potential biomarkers in psychiatric disorders, but with regard to SAD the available data is so far limited. Of note, inflammation accelerates tryptophan metabolism resulting in serotonin neurotransmission abnormalities and making anxiety reactions more sensitive [26].

With regard to neuroimaging, most of the studies on anxiety disorders focused on the amygdala, considered one of the brain areas involved in fear/anxiety and related behavioral responses [27]. Indeed, some cognitive biases among individuals with SAD, in terms of emotional hyperactivity, have been reported. Nevertheless, patients affected by SAD show impairments in different cognitive functions including verbal and visuospatial memory, verbal learning and processing speed, although some authors did not identify significant differences between patients and HC [28,29].

Taken as a whole, despite having received recent attention, the available literature on biological markers for SAD remains inconsistent and sparse. The aim of the present review is, therefore, to provide an exhaustive and comprehensive perspective on the topic.

## 2. Methods

This systematic review was performed according to the Preferred Reporting Items for Systematic Reviews and Meta-Analyses (PRISMA) guidelines [30]. A search was performed in the main psychiatric databases—PubMed (National Library of Medicine), PsychINFO, EMBASE (Ovid), Cochrane Library, etc.—to find relevant papers.

Moreover, the registries of US NIH (National Institutes of Health, ClinicalTrials.gov) clinical trials were consulted. All the original articles written in English language from 1987 to 30 June 2022, with available abstract and full texts, were included.

Two authors subsequently checked and extracted data from included articles: paper author and title, publication year, characteristics of the study: trial design, sample size, duration of study and type of biomarkers. If relevant data were not reported in the selected articles, the corresponding author was contacted to obtain further information. Disagreements were solved with the intervention of a third author who supervised the complete research activity.

The search was performed using the keywords “((social anxiety disorder) OR (social phobia)) AND ((biomarker) OR (marker))”.

Inclusion criteria were: (1) original articles; (2) mean age of patients ≥18 years; (3) reported diagnosis of SAD; (4) topic of the article focused on the association between specific biological markers and SAD.

Exclusion criteria were: (1) reviews, meta-analyses, commentaries, letters, case reports, pooled analyses, comments, case studies, study protocols; (2) mean age of the subjects under 18 years; (3) studies not providing specific results for SAD; (4) studies focusing on psychopathological dimensions other than the SAD diagnosis (e.g., markers of treatment response, markers of specific SAD symptoms); (5) studies conducted on animals; (6) articles not written in the English language. The search strategy and the inclusion and exclusion criteria followed PRISMA guidelines [31].

After data extraction from selected manuscripts, references of the selected papers were checked to identify any other potential study that did not emerge from the first search. Same inclusion and exclusion criteria were followed to select articles from references.

Quality rating was performed according to criteria by Armijo-Olivo et al. [32], and the effect sizes were calculated as Cohen’s d for all the statistically significant results of the primary objectives of the included studies.

## 3. Results and Discussion

To the best of our knowledge, this is the first systematic review including studies exploring different biological markers in SAD. We initially identified 662 papers. Among these, 508 were excluded according to the above-mentioned criteria. Therefore, fifty-four papers, together with other seven papers from references, satisfied the inclusion criteria and were included in the present systematic review (Figure 1).

Two main considerations can be made before the critical summary of the available data. First, most findings were from cross-sectional studies, examining mainly different biomarkers in people with SAD and HC. Second, according to the criteria for assessing the risk of bias (Cochrane Collaboration Risk of Bias Tool) [32], most papers were rated as of moderate quality.

In the following paragraphs, data about biological markers for SAD will be described and discussed. In Table 1, Table 2, Table 3, Table 4, Table 5, Table 6, Table 7 and Table 8, a summary of evidence according to the different groups of biomarkers is provided.

### 3.1. Genetics

All selected studies presented a moderate quality score (QS). The serotonin transporter gene (Solute Carrier Family 6 Member 4—*SCL6A4*) was studied in two papers. In the most recent one, the authors performed a genotyping of 24 single-nucleotide polymorphisms (SNPs) in 321 SAD subjects and 804 controls, extending their analysis to investigate the association to SAD severity and harm avoidance. Authors failed to report significant associations with SAD [33]. The second one included 17 families (122 first-degree family members of probands affected by SAD) and investigated the genetic link between generalized social phobia and the serotonin transporter gene (*5HTT*), or the 5HT2A receptor gene (*5HT2AR*), finding no linkage to SAD [34]. A further study, conducted on the same sample of patients, tested the linkage examining polymorphisms in dopamine receptor genes (*DRD2*, *DRD3*, *DRD4*) and the dopamine transporter gene (*DAT1*), but it failed to show linkage for all the loci [35].

The remaining two studies reported positive findings. The first one reported a linkage to SAD for chromosome 16 [36], while the other an increased risk of developing SAD for individuals carrying the C allele of the mannosidase endo-alpha gene (*MANEA*) [37]. Of note, the corresponding enzyme is involved in processes of protein glycosylation [37].

Discussion: While the specific etiology and biologic underpinnings of SAD are not well understood, family and twin studies showed a genetic vulnerability to SAD [38]. Indeed, those who have first-degree relatives with SAD have a higher likelihood and susceptibility of developing SAD [39]. Consequently, an increasing amount of studies investigated the role of genes potentially conferring vulnerability to this condition [34]. Gelernter et al. [36] found a linkage to SAD on chromosome 16 at position 71.1, hypothesizing that the most obvious candidate gene mapped in this broad region may be *SLC6A2* (“solute carrier family 6 member 2”), codifying for the noradrenaline transporter (NET). NET regulates noradrenaline and dopamine concentrations in synaptic spaces. These neurotransmitters play an important role in manifestations of anxiety, and the blockade of both norepinephrine and dopamine can be engaged in relief of some symptoms of anxiety [40]. Nevertheless, NET is a target of some medications that are effective for the treatment of SAD, including antidepressants [41]. Despite these hypotheses, the study by Kennedy et al. [35] failed to find an association between SAD and polymorphisms of the genes involved in the dopaminergic system including *DRD2*, *DRD3*, *DRD4*, *DRD4* and *DAT1*. On the other hand, Forstner et al. [33] reported a significant association between the polymorphism rs140701 at the *SLC6A4* gene and SAD, although statistical significance was lost after adjustment for multiple comparisons. *SLC6A4* encodes an integral membrane protein that transports the neurotransmitter serotonin from synaptic spaces into presynaptic neurons. This protein is the primary target of selective-serotonin reuptake inhibitors used for the treatment of anxiety disorders [42]. A dysfunction in serotonin neurotransmission was found in many psychiatric disorders, especially in affective disorders and anxiety-related phenotypes [43]. Finally, a study found an association between SAD and genetic variants of *MANEA* [37]. It was hypothesized that the different variants of *MANEA* affect the levels in the brain of the corresponding protein thus modulating the risk of anxiety disorders [44].

Globally, data about genetic biomarkers for SAD come from studies characterized by different designs and heterogeneous targets of investigation. No clear conclusions or recommendations can be drawn due to insufficient evidence and a lack of replication of positive findings. Further studies with larger samples and a prospective design are necessary to explore those genetic polymorphisms that can potentially confer vulnerability to SAD.

**Table 1 ijms-24-00835-t001:** Summary of evidence regarding genetic and epigenetic biomarkers of SAD.

BIOMARKER TYPE	REFERENCE	BIOMARKER UNDER STUDY	STUDY DESIGN	SAMPLE SIZE	FINDINGS	QUALITY SCORE ^1^	COHEN’S d
**Genetics**	[33]	*SLC6A4*	SNPs genotyping (SAD vs. HC)	1125	Two SNPs with nominal significance: - rs818702 (*p* = 0.032) - rs140701 (*p* = 0.048) After Bonferroni’s correction: SAD = HC (*p* > 0.05)	Moderate	/
	[34] ^2^	*5HTT* (*SLC6A4*), *5HT_2A_R*	Linkage analysis (first-degree family members of probands affected by SAD)	17 families (122 members)	No linkage to SAD (*p* > 0.05)	Moderate	/
	[35] ^2^	*DRD2*, *DRD3*, *DRD4*, *DAT1*	Linkage analysis (families of probands affected by SAD)	17 families (122 members)	No linkage to SAD (*p* > 0.05)	Moderate	/
	[36]	Genome	Genome-wide linkage scan (families of probands affected by Panic Disorder)	17 families (163 members)	**Linkage to SAD for chromosome 16** **(*p* = 0.0003)**	Moderate	d = 0.165
	[37]	*MANEA*	Multi-stage association study (SAD vs. HC)	131	**C allele of the *MANEA* (rs1133503*C)** **(*p* = 0.004)**	Moderate	d = 0.422
**Epigenetics**	[22]	*OTXR* methylation	Multilevel epigenetic study (SAD vs. HC)	220	**↓ *OXTR* methylation** **at CpG3 (Chr3: 8 809 437)** **(*p* < 0.001)**	Strong	d = 0.535
	[45]	Genome methylation	Epigenome-wide association study (SAD vs. HC)	143	**DMRs within *SLC43A2* (*p* < 5 × 10^−4^) and *TNXB* (*p* < 3 × 10^−26^)**	Weak	NA

Legend: ^1^ according to [32]; ^2^ analyses conducted on the same sample; **↓ =** reduced; *5HT_2A_R* = serotonin receptor 2A; *5HTT* = serotonin transporter; *DAT* = dopamine transporter; dg = diagnosis; DMRs= differentially methylated regions; *DRD* = Dopamine Receptor D; HC = healthy controls; *MANEA* = endomannosidase gene; NA = not applicable; *OXTR* = oxytocin receptor; SAD = social anxiety disorder; *SCL6A4* = Solute Carrier Family 6 Member 4; *SLC43A2* = Solute Carrier Family 43 Member 2; SNP = Single-Nucleotide Polymorphism; *TNXB* = Tenascin XB. In bold statistically significant findings.

### 3.2. Epigenetics

The authors of a recent paper analyzed whole-genome methylation of 143 SAD patients, and they found differentially methylated regions (DMRs) within intron 5 of the *SLC43A2* gene (*p* < 5 × 10^4^) and within exon 4 of the *TNXB* gene (*p* < 3 × 10^−26^) [45]. Of note, *SLC43A2* encodes the essential amino-acid transporter LAT4, while *TNXB* encodes tenascin-X, an extracellular matrix glycoprotein that is thought to play an important role in connective tissue structure. A less recent large sample study (N = 220) investigated the oxytocin receptor gene (*OXTR*) methylation and reported less OXTR methylation in SAD patients than HC [22].

Discussion: Epigenetic mechanisms alter gene expression without affecting the DNA sequence and include DNA methylation. This occurs mostly, but not exclusively, on cytosines preceding a guanine (i.e., CpG sites). Since DNA methylation changes over time according to environmental factors, this biological mechanism plays an important role in the environmental adaptation during pregnancy and childhood [46]. However, very few studies have investigated epigenetic aspects in SAD until now, although this disorder is characterized by a difficulty in facing social stimuli. Wiegand and colleagues [45] found two DMRs: one in an intron of *SLC43A2* and one in the coding region of TNXB. Changes in expression of *SLC43A2* in response to psychotherapy were reported by some authors in patients affected by anxiety disorders [47]. On the other hand, some studies suggested a role of *TNXB* in increasing the risk of psychiatric disorders [48]. Particularly, hypomethylation of a CpG site in intron 6 of *TNXB* was found to be related with panic disorder in an epigenome-wide association study [49].

*OXTR* appears to be another area of interest for studying the pathogenetic mechanisms underlying SAD, since oxytocin is important for social attachment and human bonding [50]. OXT is a neuropeptide produced in the hypothalamic paraventricular and supraoptic nuclei. It is secreted into the posterior lobe of the pituitary gland and binds to a G protein-coupled receptor widely expressed in the central nervous system. A study by Ziegler and colleagues [22] reported that decreased *OXTR* methylation was associated with the phenotype of SAD, with increased cortisol response to the Trier Social Stress Test (a stress paradigm comprising three phases), and with increased amygdala responsiveness during social phobia-related word processing. Over-expression of OXTR could reflect a compensatory up-regulation for pathologically reduced oxytocin levels in SAD and related traits [51].

To date, the paucity of epigenetic studies does not allow one to draw definitive conclusions on the possible role of epigenetic markers in the etiology of SAD. Although promising, these findings are preliminary and need further confirmation in larger samples. Nevertheless, given the dynamic regulation of gene expression through methylation, some variables, including chronic stressful life events or cortisol levels, might have affected the results of the studies [52].

### 3.3. Endocrine Biomarkers

#### 3.3.1. Cortisol

Cortisol secretion was examined in a wide range of physiological and iatrogenic conditions (e.g., unstressed, stressed, post-dexamethasone administration, etc.) and type of samples (plasma, salivary, urinary, etc.) in patients affected by SAD. Six research papers, investigating the hormone with different methodologies and under different conditions, found no significant differences in cortisol measurements between SAD patients and HC [53,54,55,56,57,58]. The oldest article by Uhde and collaborators [55] compared urinary-free cortisol (UFC) and post-dexamethasone plasma cortisol between SAD patients (UFC n = 54; cortisol n = 64) and HC, without finding statistically significant differences. Laufer and colleagues [58], in their moderate-quality study, examined cortisol levels in platelets of untreated SAD patients (N = 26) and HC (N = 21) concomitantly with other steroids (see “Sexual steroids” paragraph for results in this regard). No significant differences were observed neither in terms of cortisol plasma levels, nor of cortisol/DHEA and cortisol/DHEA-S ratios between patients and HC. All the subsequent four studies measured salivary cortisol in different conditions. For example, Klumbies and colleagues [53], evaluated plasma and salivary cortisol responses to a standardized stress situation (the Trier Social Stress Test, TSST) in 88 SAD patients and 78 HC. The authors also assessed basal long-term cortisol secretion by measuring its concentrations in scalp-hair, again without finding significant differences between patients and HC. Tamura et al. [54] also evaluated cortisol response to stress, albeit not social, by measuring salivary levels after applying physical pain in the form of electrical stimulation in 32 patients. Patients and HC showed no differences in cortisol measurements at any time point. Moreover, Van veen et al. [56] assessed salivary cortisol levels in untreated SAD patients without comorbidity and HC; measurements were performed after awakening, during the day (including late evening) and after a low dose (0.5 mg) of dexamethasone. No differences in cortisol concentrations were found in any of the above-mentioned conditions. A further research paper compared salivary cortisol secretion in a large sample of individuals divided into three groups: HC (N = 342), participants with a remitted anxiety disorder (N = 311; SAD n = 140) and participants with a current anxiety disorder (N = 774; SAD n = 487). Salivary cortisol was measured during the first hour after awakening (“1-h awakening response”), in the evening and after dexamethasone challenge. Although having a current anxiety disorder was associated with higher awakening cortisol levels, these findings were only significant for patients with panic disorder with agoraphobia and those with comorbid depression [57].

**Table 2 ijms-24-00835-t002:** Summary of findings regarding endocrine biomarkers of SAD.

HORMONAL SYSTEM	REFERENCE	BIOMARKER UNDER STUDY	STUDY DESIGN	SAMPLE SIZE	FINDINGS	QUALITY SCORE ^1^	COHEN’S d
**Cortisol and sAA**	[53]	Salivary cortisol, plasma cortisol, sAA, prolactin	Cross-sectional (SAD vs. HC) Secretion after TSST	166	SAD = HC: - Salivary cortisol stress response (*p* > 0.131) - Plasma cortisol stress response (*p* = 0.084) - sAA stress response (*p* > 0.343) - Prolactin stress response (F < 1) - Basal hair cortisol levels (*p* = 0.918)	Moderate	/
	[54]	sAA, salivary cortisol	Cross sectional (SAD vs. HC) Response to electrical stimulation	112	**sAA: SAD > HC at all-time points (*p* < 0.01)** Salivary cortisol: SAD = HC at all-time points (*p* > 0.05)	Moderate	sAA: d = 0.576
	[55]	UFC, post-dexamethasone plasma cortisol	Cross-sectional (SAD vs. HC)	54 patients (UFC); 64 patients (plasma cortisol)	SAD = HC - UFC (*p* = 0.15) - Post-dexamethasone (*p* = 0.37)	Weak	/
	[56]	sAA, salivary cortisol Measured in non-stressed conditions: - after awakening - during the day (including late evening) - after a low dose (0.5 mg) of dexamethasone	Cross-sectional (SAD vs. HC)	86	SAD = HC - Awakening sAA (*p* = 0.114) - Salivary cortisol (awakening, diurnal, late evening) (*p* = 0.201) - Post-dexamethasone salivary cortisol (*p* = 0.256) SAD > HC: **- Diurnal sAA (*p* = 0.044)** **- Post-dexamethasone sAA (*p* = 0.040)**	Moderate	Diurnal sAA: d = 0.508 Post-dexamethasone sAA: d = 0.518
	[57]	Salivary cortisol - 1-h cortisol awakening response - evening cortisol - cortisol response after 0.5 mg of dexamethasone ingestion	Cross sectional (ADs vs. HC)	Current: 140 SAD patients Remitted: 487 SAD patients	SAD = HC (*p* > 0.05)	Moderate	/
	[59]	Plasma ACTH, plasma cortisol, salivary cortisol	Cross-sectional (SAD vs. HC) Secretion after TSST	70	Baseline ACTH, plasma and salivary cortisol: SAD = HC (*p* > 0.05) ACTH secretion pattern and AUCg: SAD = HC (*p* > 0.05) **Plasma cortisol: SAD < HC** **- secretion pattern (*p* = 0.011)** **- AUCg (*p* = 0.007)** Salivary cortisol: - secretion pattern SAD = HC (*p* > 0.05); **- AUCg, SAD < HC (*p* = 0.007)**	Strong	Plasma cortisol: d = 0.165 (secretion pattern); d = 0.229 (AUCg) Salivary cortisol: d = 0.229 (AUCg)
	[60]	Plasma cortisol, plasma ACTH	Cross-sectional (SAD vs. HC) Secretion after stressors: - mental arithmetic - short-term memory test in front of an audience	30	Baseline ACTH and cortisol: SAD = HC (*p* > 0.05) **Delta max ^2^ cortisol response: SAD > HC (*p* < 0.04)** Delta max ^2^ ACTH response: SAD = HC (*p* > 0.05)	Strong	Delta max ^2^ cortisol response: d = 0.767
	[61]	Plasma cortisol and EA relationship	Cross-sectional (SAD vs. HC) Secretion after TSST	24	**SAD ≠ HC (*p* = 0.015)** **- SAD: ↓ total cortisol secretion (AUCg) with ↑ EA score (r2 = −0.177)** **- HC: ↑ total cortisol secretion (AUCg) with ↑ EA score (r2 = 0.49)**	Weak	d = 2.606
	[62]	Cortisol plasma level and 5-HT_1A_ receptor distribution	Cross-sectional (SAD vs. HC)	30	**Plasma cortisol: SAD < HC** **(*p* = 0.016)**	Moderate	d = 0.897
	[63]	sAA	RCT sAA levels after TSST	39	SAD = HC (*p* > 0.05)	Strong	/
**Sexual steroids and cortisol**	[58]	Platelet DHEA, DHEA-S, pregnenolone and cortisol	Cross-sectional (SAD vs. HC)	47	SAD = HC - DHEA: *p* = 0.75 - DHEA-S: *p* = 0.490 - Pregnenolone: *p* = 0.500 - Cortisol: *p* = 0.1285 - Cortisol/DHEA: *p* = 0.18 - Cortisol/DHEA-S: *p* = 0.72	Moderate	/
**Sexual steroids**	[64]	Salivary testosterone	Cross-sectional (ADs vs. HC)	SAD patients: 135 males and 252 females	**SAD females < HC (*p* < 0.001)** SAD males = HC (*p* = 0.76)	Moderate	Females: d = 0.299
	[65]	PREG-S, ALLO and DHEA-S	Cross-sectional (SAD vs. HC)	24 males	PREG-S: SAD < HC (*p* = 0.008) ALLO: SAD = HC (*p* = 0.96) DHEA-S: SAD = HC (*p* = 0.165)	Moderate	PREG-S: d = 1.184
**Thyroid hormones**	[66]	T3, T4, fT4, TSH, anti-thyroid Ab	Cross-sectional (SAD vs. HC)	43 patients	SAD = HC - T3: *p* = 0.59 - T4: *p* = 0.95 - fT4: *p* = 0.81 - TSH: *p* = 0.81 - TSH TRH response pattern: *p* = 0.71 - Antityhroid Ab: *p* > 0.05	Moderate	/

Legend: ^1^ according to [32]; ^2^ the difference between baseline values and the maximum increase during the stressor; **↓ =** reduced; ↑ = increased; ACTH = Adrenocorticotropic hormone; ADs = Anxiety Disorders; ALLO = allopregnanolone; AUCg = area under the curve ground; DHEA-S = dehydroepiandrosterone sulphate; EA = emotional abuse; fT4 = free thyroxine; HC = healthy controls; PREG-S = pregnenolone-sulphate; RCT = randomized control trial; sAA = salivar alpha amylase; SAD = social anxiety disorder; T3 = triiodothyronine; T4 = thyroxine; TRH = thyroid Releasing Hormone; TSH = Thyroid Stimulating Hormone; TSST = Tier Social Stress Test; UFC = urinary free cortisol. In bold statistically significant findings.

Two studies, characterized by a high QS, yielded positive, yet opposite, findings. Recent research found decreased plasma cortisol secretion after TSST administration in SAD patients compared to HC [59]. On the other hand, a less recent paper, measuring plasma cortisol levels after a similar social stressor, reported increased cortisol in SAD patients [60].

Given the emerging evidence that early childhood adversity may affect cortisol reactivity in anxiety disorders, the relationship between the cortisol stress response and childhood trauma (and its subcomponent Emotional Abuse—EA) was examined by Vaccarino and colleagues [61]. The study compared 11 SAD patients and 10 HC. The main result was a significant group–EA interaction in predicting cortisol secretion, an effect driven by a very strong positive relationship between EA scores and total cortisol secretion in the HC group, and a moderately strong negative relationship between EA scores and total cortisol secretion in the SAD group [61].

Finally, Lanzerberger and collaborators [62] investigated the correlation between cortisol plasma levels and the 5-HT1A receptor distribution using Positron Emission Tomography (PET) (see “Neuroimaging” paragraph for the results). The study included a small sample size (N = 30 males) and presented a moderate QS. With regards to hormonal measurements, plasma cortisol levels resulted to be significantly lower in SAD patients as compared to HC [62].

#### 3.3.2. Salivary Alpha Amylase

Salivary alpha amylase (sAA), which is locally secreted in the oral mucosa of salivary glands, represents a quick and sensitive biological marker of the Autonomic Nervous System activity and of stress response. Four papers measured sAA in SAD, with contradictory results. One moderate-QS observational study [53] and one high-QS randomized controlled trial (RCT) [63], evaluating sAA levels after TSST administration, found no significant differences between SAD and HC. One moderate-QS paper, measuring sAA after electrical stimulation, reported a significant increase of sAA levels in SAD patients with respect to HC [54]. Finally, a moderate-QS paper evaluating sAA levels after awakening, during the day, and after 0.5 mg of dexamethasone, found significantly increased sAA in diurnal and post-dexamethasone levels in patients than HC [56].

#### 3.3.3. Sexual Steroids

A moderate-quality study with a relatively large sample of SAD patients (N = 135 males and 252 females) found decreased salivary testosterone levels in female patients compared to HC [64]. One moderate-QS study with a small sample size (N = 24 males), evaluating plasma pregnenolone sulphate (PREG-S), allopregnanolone (ALLO) and dehydroepiandrosterone sulphate (DHEA-S), found decreased PREG-S levels in SAD patients with respect to HC, and no differences for the other neurosteroids [65]. Finally, a moderate-QS article with a small sample size (N = 47) failed to identify significant differences between patients and HC in plasma levels of DHEA, DHEA-S and PREG [58].

A role of neuroactive steroids in the modulation of the gamma-aminobutyric acid type A/benzodiazepine receptor complex has been highlighted, suggesting that these steroids may exert an anxiolytic or anxiogenic activity depending on their receptor positive (e.g., allopregnanolone) or negative allosteric modulation (e.g., dehydroepiandrosterone sulphate) [67,68].

#### 3.3.4. Thyroid Hormones

One article with a moderate QS and a small-to-medium sample size (N = 43 patients) evaluated thyroid-related hormones and antibodies, and no significant differences were identified between subjects affected by SAD and HC [66].

Discussion: Most studies focused on cortisol as a potential endocrine biomarker associated with SAD. Cortisol is commonly used as a stress biomarker given that it activates metabolism and “fight or flight” behavioral responses [69]. Indeed, prolonged exposure to stress causes changes in the body, such as an activation of the hypothalamic–pituitary–adrenal axis (HPA), which results in an elevated secretion of cortisol [69]. Chronic hypercortisolemia may increase the vulnerability to mental disorders such as depression and anxiety disorders [57]. Studies investigating HPA axis function in anxiety disorders usually measured cortisol secretion as an indicator of HPA activation and compared patients with anxiety disorders to control groups [70]. The available literature shows inconsistent results: on one side, some evidence points to higher cortisol secretion in people with anxiety disorders compared to controls [71]. On the other hand, some studies found comparable cortisol levels between the two groups [25]. It was suggested that patients with SAD would have an initial HPA axis activation and increased cortisol reactivity as a result of social exposition, but that these biological responses would attenuate in case of chronic exposure to stressful situations [61]. This model could explain the contrasting results reported for patients affected by SAD: Petrowski et al. [59] reported HPA axis hypo-reactivity, opposite results were found by Condren et al. [60], while six studies [53,54,55,56,57,58] failed to find significant differences in cortisol levels between SAD patients and HC. One confounding factor can be represented by age at onset, because a childhood onset of anxiety disorders is associated with a reduced basal HPA tone, attenuated parasympathetic and increased sympathetic activity [72,73].

With regard to the other hormones, contrasting findings were reported in SAD patients for neurosteroids [58,65], while salivary testosterone levels resulted to be lower in female patients compared to controls [64]. Decreased PREGS concentrations in SAD patients versus HC [65] might represent an insufficient compensatory mechanism, given that PREGS is anxiolytic at low levels, but anxiogenic at high levels [74]. On the other hand, low levels of testosterone may represent an imbalance between testosterone and cortisol. Of note, androgens downregulate the HPA axis in a direct way by androgen receptors which are expressed in corticotropin-releasing hormone (CRH) neurons in the hypothalamus and facilitate the inactivation of the promoter region of the CRH gene [75].

Finally, sAA can represent a biomarker of the activation of the sympathetic adrenomedullary (SAM) system. The SAM system, through catecholamine signaling and interacting with the HPA system, plays an important role in both normal homeostasis and in stress responses [76]. However, the results in SAD patients are controversial also for this biomarker [53,54,56,63].

Taken as a whole, the findings about endocrine factors as biomarkers of SAD are contrasting and may have been influenced by the heterogeneity of study designs, the lack of standardized laboratory kits and different clinical severity of enrolled participants [77].

### 3.4. Immunological Markers

Among the seven included studies, four papers found no significant findings, whilst three reported positive results. CRP and several cytokines, particularly Interleukin-6 (IL-6) and Tumor Necrosis Factor-α (TNF-α), were the most investigated markers.

**Table 3 ijms-24-00835-t003:** Summary of findings regarding immunological biomarkers of SAD.

REFERENCE	BIOMARKER UNDER STUDY	STUDY DESIGN	SAMPLE SIZE	FINDINGS	QUALITY SCORE ^1^	COHEN’S d
[78]	IL-6, TNF-α, hsCRP	Prospective (ADs vs. HC)	384 SAD patients: current = 255, remitted = 129	SAD = HC Current: - IL-6 *p* = 0.963 - TNF-α *p* = 0.314 - hsCRP *p* = 0.129 Remitted: - IL-6 *p* = 0.241 - TNF-α *p* = 0.925 - hsCRP *p* = 0.365	Moderate	/
[79]	CRP	Cross-sectional (ADs vs. HC)	508 SAD patients	SAD = HC (*p* = 0.124)	Moderate	/
[80]	CRP, IL-6, TNF-α	Cross-sectional (ADs vs. HC; comparison among ADs)	651 SAD patients	ADs = HC (*p* > 0.05) **CRP: SAD < ADs (*p* = 0.04)** **IL-6:** **SAD < ADs (*p* = 0.05);** **female SAD patients *p* = 0.007; male SAD patients *p* = 0.61** TNF-α: SAD = ADs (*p* = 0.64)	Moderate	CRP: d = 0.207 IL6: d = 0.205 (female SAD patients d = 0.281)
[81]	LPS-stimulated TNF-α, IL-6, IL-8	Cross-sectional (SAD vs. HC; OCD vs. HC; SAD vs. OCD)	26 SAD patients	SAD = HC TNF-α: *p* = 0.69 IL-6: *p* = 0.82 IL-8: *p* = 0.62 SAD = OCD TNF-α: *p* = 0.971 IL-6: *p* = 0.076 IL-8: *p* = 0.103	Weak	/
[82]	Basal and LPS-stimulated IFN-γ, IL-10, IL-1β, IL-2, IL-4, IL-6, IL-8, TNF-α	Cross-sectional (SAD vs. HC)	68	Stimulated IFN-γ, IL-10, IL-1β, IL-2, IL-4, IL-6, IL-8, TNF-α: SAD = HC (*p* > 0.05) Unstimulated IFN-γ, IL-1β, IL-2, IL-4, IL-6, IL-8, TNF-α: SAD = HC (*p* > 0.05) **Unstimulated IL-10:** **- SAD < HC (*p* = 0.04)** **- SAD males < HC (*p* = 0.016)** - SAD females = HC (*p* = 0.30)	Moderate	Unstimulated IL-10: d = 0.517 (males: d = 0.872)
[83]	IL-2, soluble IL-2R	Cross-sectional (SAD vs. HC)	30	SAD = HC (*p* > 0.05)	Weak	/
[84]	Circulating lymphocyte phenotypic surface markers	Cross-sectional (SAD vs. HC; PD vs. HC)	65	**↑ CD16 (*p* < 0.05)**	Moderate	d = 0.442

Legend: ^1^ according to [32]; ↑ = increased; ADs = Anxiety Disorders; CRP = C-Reactive Protein; HC = Healthy Controls; hsCRP = high sensitivity C-Reactive Protein; IL = Interleukin; KYN = kynurenine; KYNA = kynurenic acid; LPS = lipopolysaccharide; OCD = Obsessive-Compulsive Disorder; SAD = Social Anxiety Disorder; TNF-α = Tumor Necrosis Factor—alpha; TRYP = tryptophan. In bold statistically significant findings.

CRP was examined by three different studies: two were moderate-QS studies [78,79] with fairly large samples of SAD patients (N = 384; N = 508, respectively). The first one, by Glaus and colleagues [78], involved subjects from a Swiss civil register and had the objective to examine inflammatory markers at baseline and after a mean follow-up period of 5.5 (±0.6) years. The authors found no significant associations between IL-6, TNF-α and high-sensitivity CRP (hsCRP) with current or remitted social phobia. The second one, by Naudé and collaborators [79], also reported negative findings regarding the association of serum CRP with SAD. A further study, with greater statistical power (N = 651) found significantly lower CRP levels in patients affected by SAD compared to other anxiety disorders. The same study also demonstrated decreased IL-6 levels in SAD patients when compared to other anxiety disorders [80]. The remaining studies consistently failed to find associations between IL-6 or TNF-α and SAD [81,82]. Nevertheless, the recent work by Butler and colleagues [82] investigated different inflammatory cytokines under stimulated and unstimulated conditions, reporting decreased levels of unstimulated IL-10 in SAD patients compared to HC; however, when controlling for gender, this association remained statistically significant only for males (*p* = 0.016). No differences were found in basal or lipopolysaccharide-stimulated pro-inflammatory cytokine levels between patients and HC [82].

Moreover, Rapaport and Stein [83] explored the role of IL-2 and its soluble receptor, SIL-2R, in social phobia; patients and HC showed no statistically significant differences in mean serum levels. Finally, an article with a moderate QS reported a significant increase of CD16+ (natural killer) cells in individuals with SAD than in HC; to our knowledge, this is the only available study examining lymphocyte surface markers in SAD [84].

Discussion: Currently, the relationship between inflammation and mental illness has not been totally clarified, as there exist several proposed models [85]. Despite the increasing interest in the role of inflammation in mental disorders, few research focused on the possible relationship between the immune system and anxiety disorders [86]. Even though the existing data provide some evidence that generalized anxiety disorder and panic disorder may be associated with a pro-inflammatory state, the available literature is controversial regarding SAD. Particularly, one longitudinal [78] and one cross-sectional [79] study, both including large sample sizes, found no significant association between pro-inflammatory molecules and social phobia. Nevertheless, other three cross-sectional studies with small sample sizes [81,82,83] failed to find associations between different pro-inflammatory cytokines and SAD. Interestingly, Vogelzangs and co-authors [80] found lower CRP and IL-6 levels in a large sample of individuals with SAD compared to subjects affected by other anxiety disorders. In trying to explain such results, the authors hypothesized an effect of the age at the onset of the different anxiety disorders. In particular, SAD is associated with an earlier age at onset and may be characterized by a lower grade of inflammation than other anxiety disorders [87]. Some authors suggested the existence of a specific late-onset subtype of anxiety disorders with a distinct etiology, potentially in comorbidity with depression [88]. Even though patients with SAD do not show any increase of circulating pro-inflammatory cytokines, Butler and colleagues [82] reported lower levels of anti-inflammatory cytokine IL-10 among SAD males compared with HC. IL-10 is an important anti-inflammatory cytokine, playing a regulatory role in suppressing pro-inflammatory response. Preliminary findings showed a reduction in the release of this molecule in depression, generalized anxiety disorder and panic disorder [89] as a compensatory mechanism of excessive cortisol release and of up-regulation in the kynurenine pathway [90] caused by an over-activation of the sympathetic system (that facilitates the production of inflammatory cytokines by immune cells) [63,91]. In that regard, Rappaport [77] found that both subjects with panic disorder and those with social phobia had increased NK-cell numbers when compared with healthy volunteers. An increased release of catecholamines cause a transient redistribution of lymphocytes from the peripheral vasculature to the central circulation [91] and could be responsible for the rise of NK-cells in patients affected by SAD.

In light of the heterogeneity among studies, different sample sizes and confounding factors including the use of psychotropic medications [82], the findings concerning immunological markers in SAD should be interpreted with caution. It cannot be ruled out that immune dysregulation is not a general phenomenon in anxiety disorders, but it might be restricted to specific subgroups identified by gender, age at onset, type of disorder, duration and severity of the disorder, or its comorbidity with psychiatric or non-psychiatric disorders [92].

### 3.5. Antioxidant Markers

Butler and colleagues [82], previously cited for findings regarding IL-10, also investigated the immune-kynurenine system in a small sample of SAD subjects (N = 32), in light of the emerging evidence that the tryptophan-kynurenine (TRYP—KYN) pathway may be altered in several psychiatric disorders and seems to be associated with suicidal behavior [93]. SAD patients presented higher plasma kynurenic acid (KYNA) levels and an increased KYNA/KYN ratio compared to HC, without differences in KYN, TRYP or the KYN/TRYP ratio. Moreover, individuals affected by SAD with lifetime suicide attempts showed higher plasma KYN levels and an elevated KYN/TRYP ratio with respect to subjects without a history of suicide attempts [82].

**Table 4 ijms-24-00835-t004:** Summary of findings regarding antioxidant system biomarkers of SAD.

REFERENCE	BIOMARKER UNDER STUDY	STUDY DESIGN	SAMPLE SIZE	FINDINGS	QUALITY SCORE ^1^	COHEN’S d
[82]	KYN, TRYP, KYNA, KYN/TRYP, KYNA/KYN	Cross-sectional (SAD vs. HC)	68	**SAD > HC** **- KYNA (*p* = 0.0005)** **- KYNA/KYN (*p* = 0.0005)** KYN: *p* = 0.78 TRYP: *p* = 0.58 KYN/TRYP: *p* = 0.70	Moderate	KYNA: d = 0.941 KYNA/KYN: d = 1.037
[94]	SOD, CAT, GSHPx, MDA	Clinical Trial (SAD vs. HC; pre- and post- citalopram treatment)	78	**Baseline: SAD > HC** - SOD (*p* < 0.05) - CAT (*p* < 0.01) - GSH-Px (*p* < 0.01) - MDA (*p* < 0.01) All ↓ after citalopram treatment (*p* < 0.05)	Weak	Baseline SOD: d = 1.297 Baseline CAT: d = 1.352 Baseline GSH-Px: d = 1.5827 Baseline MDA: d = 2.0111
[95]	SOD, GSH-Px, CAT, MDA	Cross-sectional (SAD vs. HC)	36	**SAD > HC** - SOD (*p* < 0.01) - CAT (*p* < 0.01) - GSHPx (*p* < 0.001) - MDA (*p* < 0.001)	Weak	SOD: d = 1.319 CAT: d = 0.891 GSHPx: d = 1.816 MDA: d = 4.859
[96]	AGEs	Cross-sectional case-control (affective disorders and ADs vs. HC)	691 SAD patients	SAD = HC (*p* > 0.05)	Moderate	/

Legend: ^1^ according to [32]; ADs = anxiety disorders; AGEs = Advanced Glycation End products; CAT = catalase; GSHPx = glutathione peroxidase; HC = healthy controls; KYN = kynurenine; KYNA = kynurenic acid; MDA = malondialdehyde; SAD = social anxiety disorder; SOD = superoxide dismutase; TRYP = tryptophan. In bold statistically significant findings.

Two weak-QS studies with small sample sizes, conducted by the same research group, found elevated levels of antioxidant enzymes (superoxide dismutase—SOD, glutathione peroxidase—GSH-Px and catalase—CAT) and malondialdehyde (MDA), a lipid peroxidation product, in SAD patients with respect to HC [94,95]. In particular, the first one was a clinical trial conducted on 78 individuals (39 affected by SAD) comparing the levels of antioxidant enzymes and MDA between patients and HC, and before and after an 8-week period of citalopram treatment. The authors found a significant difference at the baseline on all the antioxidant system markers between patients and HC, and a statistically significant decrease of all enzymes and MDA after citalopram treatment, leading authors to conclude that antioxidant system enzymes and MDA may be considered a state marker of social phobia [94]. The other study, using a cross-sectional design, demonstrated elevated antioxidant enzymes and MDA levels in 18 SAD patients when compared to 18 HC, and also found positive correlations between each parameter and severity of social phobia, and between the duration of illness and MDA, SOD or CAT [95].

The last study, with a moderate QS, examined end-glycation products in a large sample of individuals with SAD (N = 691). Advanced Glycation End products (AGEs) are considered stable (trait) biomarkers of cumulative oxidative stress being produced through non-enzymatic glycation and oxidation of proteins or their degradation products. Hagen and colleagues [96] did not find significant differences between SAD patients and HC.

Discussion: The KYN pathway, the major route for TRYP catabolism, has been largely studied for different mental disorders as it is one of the biological systems explaining the connection between neurotransmission abnormalities and inflammation [93,97]. This pathway is active both in periphery and in the CNS. Following the conversion of TRYP to N-formylkynurenine by indoleamine 2,3- dioxygenase 1 (IDO) or tryptophan 2,3-dioxygenase (TDO), the resulting molecules are degraded to KYN, that is a precursor of neuroactive compounds as quinolinic acid (QUIN) and KYNA, both affecting glutamatergic neurotransmission. QUIN exhibits neurotoxic properties since it inhibits astrocytic uptake of glutamate and induces neuronal glutamate release in addition to being one of the neuroactive metabolites produced by microglia and infiltrating macrophages [98]. On the other hand, KYNA reduces extracellular brain glutamate levels, avoids the release of inflammatory cytokines and acts as an antioxidant able to scavenge free radicals [99]. Recent reports indicate that the activation of the KYN pathway, caused by an activation of pro-inflammatory factors accompanying neurodegenerative processes, leads to the accumulation of its neuroactive and pro-oxidative metabolites [99]. However, the Central Nervous System is particularly vulnerable to oxidative stress for different reasons including the high request of oxygen resulting in excessive reactive oxygen species production, and the fact that neuronal membranes are rich in polyunsaturated fatty acids that are easily attacked by free radicals. In this regard, lipid peroxidation of neuronal membranes in turn negatively impacts on signal transduction, synaptic plasticity, mitochondrial function and ultimately on neuronal survival [100]. One of the final products of lipid peroxidation is Malondialdehyde (MDA), that represents a peripheral biomarker of oxidative stress of CNS and of lipid damage of neuronal membranes [101]. The degree of oxidative stress seems to be proportional to the severity of symptoms of many psychiatric and neurodegenerative disorders [102]. However, with regard to SAD, results are inconsistent. Two studies reported a possible relationship between a diagnosis of SAD and both antioxidant enzymes and MDA [94,95]. Antidepressants seem to reduce the antioxidant enzyme and MDA levels in subjects affected by SAD [94,103]. On the other hand, Hagen and colleagues [96] failed to find a significant association between oxidative stress (namely AGEs, the production of which is highly dependent upon the degree of oxidative stress) and SAD in a sample including 691 patients. Nevertheless, the same authors found a relationship between AGEs and affective disorders globally, independently from sociodemographic and medical comorbidities. Finally, Butler and co-authors [82] found increased plasma levels of KYNA as well as an elevated KYNA/KYN in patients with SAD. In addition, SAD patients taking psychotropic medication had higher KYN levels than those who were medication free, suggesting a possible link between elevated KYN levels and severity of symptoms. Such findings seem to be consistent with some preliminary data which are suggestive of KYN pathway activation in anxiety states [26]. Particularly, KYN pathway activation appears to be preferably directed towards KYNA synthesis. In light of the results coming from animal models, it was hypothesized that in SAD, the chronic stress to which patients are exposed may not only increase the conversion of TRYP to KIN through TDO activation but shift downstream metabolism by the production of KYNA [104]. Other mechanisms explaining the increased KYNA plasma levels in SAD individuals may include an over-activation of the sympathetic nervous system and a gut dysbiosis affecting the TRYP-KYN pathway [105]. Nevertheless, SAD patients with a history of suicide attempts have raised KYN levels and an increased KYN/TRYP ratio compared to those without a history of suicidal attempts, as previously reported in some studies including patients with depression [106].

To date, the possible association between SAD and oxidative stress should be considered preliminary and controversial. Indeed, most studies on anxiety disorders and antioxidant system biomarkers were conducted on animal models [107]. Further prospective studies with large samples are required to confirm the described preliminary results.

### 3.6. Neurotrophic Factors

A first strong-QS paper, involving 393 unmedicated non-depressed subjects suffering from anxiety disorders (105 with SAD), did not show differences in serum levels of the Brain-Derived Neurotrophic Factor (BDNF) between patients and HC [108]. Moreover, the authors investigated sex differences in BDNF serum levels, finding that female patients had lower levels of BDNF than female HC, independently from the type of anxiety disorder (results not specified for SAD). Another paper characterized by a moderate QS, reported negative results about the difference in BDNF serum levels between 20 social phobic patients and HC [109]. Finally, Pedrotti Moreira and colleagues [110] conducted a moderate-QS study (N = 70) investigating serum Glial Cell Line-Derived Neurotrophic Factor (GDNF) levels; the authors found increased levels in SAD compared to HC.

**Table 5 ijms-24-00835-t005:** Summary of findings regarding neurotrophic markers of SAD.

REFERENCE	BIOMARKER UNDER STUDY	STUDY DESIGN	SAMPLE SIZE	FINDINGS	QUALITY SCORE ^1^	COHEN’S d
[108]	Serum BDNF	Cross-sectional (ADs vs. HC)	105 SAD patients	SAD = HC (*p* > 0.05)	Strong	/
[109]	Serum BDNF	Cross-sectional (ADs vs. HC)	20 SAD patients	SAD = HC (*p* = 0.913)	Moderate	/
[110]	Serum GDNF	Cross-sectional (ADs vs. HC)	70 SAD patients	**SAD > HC (*p* = 0.004)**	Moderate	NA

Legend: ^1^ according to [32]; ADs = anxiety disorders; BDNF = Brain-Derived Neurotrophic Factors; GDNF = Glial cell line-Derived Neurotrophic Factor; HC = healthy controls; NA = not applicable; SAD = social anxiety disorder. In bold statistically significant findings.

Discussion: Despite that BDNF and GDNF are intimately related to neuroplasticity and largely studied in animal models of anxiety [111], only in the last years these factors have been investigated in patients with anxiety disorders. Nevertheless, while the relationship between BDNF and depression is established, the role of BDNF in anxiety is less clear with some studies suggesting an association with reduced BDNF in different regions of the brain concomitantly to increased anxiety, but with other reports that fail to find such relationship [112]. In the same way, no association arose between neurotrophic factors and SAD [108,109]. Nevertheless, serum BDNF levels may be similar across the different types of anxiety disorders and thus peripheral BDNF levels do not have the specificity to categorize anxiety disorders as well as the severity of anxiety or a more chronic course [108]. Some authors made the hypothesis that low BDNF levels might be specifically related to the pathophysiology of anxiety in females [108,109]. On the other hand, even though some studies reported a possible involvement of GDNF in several psychiatric disorders, including anxiety ones, to date there are few and contradictory data to draw definitive conclusions [110].

Even though neurotrophic factors play a key role in development, synaptogenesis, differentiation and survival of neurons in the brain as well as in the adaptation to external influences and their potential role as markers of SAD require more studies.

### 3.7. Neuroimaging

Three articles investigated Resting State Functional Connectivity (RSFC), as measured by Functional Magnetic Resonance Imaging (fMRI). Among these, one moderate-QS study [113] with a limited sample size (N = 28) found that increased RSFC in the left and right amygdalae, in the temporal voice area (TVA) and between the amygdala and the TVA correlated with social anxiety severity. Moreover, Ding and colleagues [114], in another study with moderate QS and a small sample size (N = 36), found decreased positive connectivity in the frontal lobe and decreased negative connectivity between the frontal and occipital lobes in SAD. Finally, a further moderate-QS study with a larger sample size (N = 84) found 49 decreased positive connections involving the frontal, occipital, parietal–(pre) motor and temporal regions [115].

**Table 6 ijms-24-00835-t006:** Summary of findings regarding neuroimaging biomarkers of SAD.

IMAGING TECHNIQUE	REFERENCE	BIOMARKER UNDER STUDY	STUDY DESIGN	SAMPLE SIZE	FINDINGS	QUALITY SCORE ^1^	COHEN’S d
**fMRI**	[113]	RSFC	Cross-sectional (SAD vs. HC)	28	Increased LSAS score = ↑ RSFC in: - left ↔ right amygdala (*p* = 0.01) - left amygdala ↔ right TVA (*p* = 0.04) - left ↔ right TVA (*p* = 0.03)	Moderate	/
	[114]	RSFC	Cross-sectional (SAD vs. HC)	36	**↓ positive connections within the frontal lobe: right median PFC ↔ right inferior frontal cortex (*p* < 0.05)** **↓ negative connections frontal ↔ occipital lobes: right median PFC ↔ left calcarine fissure, left superior occipital cortex, left cuneus (*p* < 0.05)**	Moderate	NA
	[115]	RSFC, network topology	Cross-sectional (SAD vs. HC)	84	**- ↓ 49 positive connections (*p* < 0.05) in the frontal, occipital, parietal–(pre) motor, and temporal regions** **- ↓ default mode network connectivity (*p* < 0.01)** **- ↑ Lp (*p* < 0.01); ↓ Cp (*p* < 0.01)**	Moderate	NA
	[116]	Amygdala response to angry schematic faces	Cross sectional (SAD vs. HC)	22	**Angry vs. neutral faces:** **↑ responses (all *p* < 0.001) in:** **-- right dorsal amygdala;** **-- left supramarginal gyrus;** **-- left supplementary eye field;** **↓ right dACC response**	Moderate	- Amygdala: d = 1.782 - Left supramarginal gyrus: d = 2.029 - Left supplementary eye field: d = 1.949 - Right dACC: d = 1.914
	[117]	Amygdala activation to facial expressions	Cross-sectional (SAD vs. HC) Harsh (angry, fearful, and contemp- tous) vs. accepting (happy) facial emotional expressions	30	**↑ left allocortex activation (amygdala, uncus, parahippocampampal gyrus activation):** **- to contemptuous vs. happy faces (*p* = 0.004)** **- to angry vs. happy faces (*p* = 0.02)**	Moderate	d = 1.14 d = 1.00
	[118]	Amygdala reactivity to emotional faces with low, moderate, and high intensity	Cross sectional (SAD vs. HC)	22	**↑ left amygdala activation to high intensity emotional faces (*p* < 0.05)**	Moderate	d = 0.899
	[119]	Amygdala reactivity to threatening faces of low, moderate, and high intensity	Cross-sectional (SAD vs. HC)	24	**- ↑ left amygdala reactivity to threatening faces at moderate (*p* < 0.03) and high intensity (*p* < 0.003)** **- ↑ right amygdala reactivity for high intensity (*p* < 0.04)** SAD = HC: - left amygdala low intensity (*p* = 0.10) - right amygdala for moderate (*p* = 0.18) or low (*p* = 0.34) intensity	Moderate	- Left amygdala, moderate intensity: d = 0.947 - Left amygdala, high intensity: d = 1.361 - Right amygdala, high intensity: d = 0.891
	[120]	Amygdala connectivity to PFC at rest and during threat processing	Cross-sectional (SAD vs. HC) (EFMT; a resting state Task)	37	**At rest: ↓ connectivity** **- right amygdala ↔ rostral ACC (*p* < 0.05)** **During threat: ↓ connectivity** **- right amygdala ↔ rostral ACC (*p* < 0.05)** **- left amygdala ↔ rostral ACC (*p* < 0.05)** **- right amygdala ↔ DLPFC (*p* < 0.05)** **- left amygdala ↔ DLPFC (*p* < 0.05)**	Moderate	At rest: d= 0.557–1.649 During threat: - d = 0.557 - d = NA - d = NA - d = 0.557
	[121]	DLPFC activation during perception of laughter	Cross- sectional (SAD vs. HC)	26	**↑ activation during reappraisal in the left DLPFC (*p* = 0.007)**	Moderate	/
	[122]	Insula Reactivity and Connectivity to ACC when processing threat	Cross-sectional (SAD vs. HC) (EFMT fear, angry, happy expressions)	55	**↑ activation to fear (>happy) faces in the left aINS (*p* < 0.003) and right aINS (*p* < 0.005)** **↓ connectivity right aINS ↔ dACC during fearful face processing (*p* < 0.05)**	Moderate	- Left aINS: d = 1.2365 - Right aINS: d = 0.823 - Right aINS ↔ dACC: d = 1.4741
	[123]	Brain activation during exposure to emotional faces	Cross-sectional (SAD vs. HC)	46	**Higher LSAS scores = ↑ activation** **- left anterior insula (*p* < 0.05)** **- right lateral PFC (*p* < 0.05)**	Moderate	- Left aINS: d = 1.435 - Right lateral PFC: d = 1.666
	[124]	FC during face processing	Cross sectional (SAD vs. HC; SAD vs. PD)	Primary sample: 16 SAD patients Replication sample: 14 SAD patients	**↓ FC left temporal pole ↔ left hippocampus (*p* = 0.042), particularly with angry faces (*p* = 0.027)**	Strong	- All faces: d = 0.190 - Angry faces: d = 0.245
	[125]	Fearful face processing brain signal (whole brain, fear network parietal lobe); regional gray matter volume	Cross-sectional (SAD vs. HC) (fMRI/sMRI + SVM)	26 males	Fearful face processing: **- Whole brain activation: SAD ≠ HC (*p* = 0.034)** **- Fear network activation: SAD ≠ HC (*p* = 0.017)** - Parietal lobe activation: SAD = HC (*p* = 0.548) Gray matter volume: **- Whole brain: SAD ≠ HC (*p* = 0.001)** - Regional in fear network: SAD = HC (*p* = 0.397) - Regional in parietal lobe: SAD = HC (*p* = 0.232)	Moderate	- Whole brain activation: d = 0.742 - Fear network activation: d = 0.954 - Whole brain gray matter volume: d = 1.896
	[126]	CT and CSA	Cross sectional (SAD vs. HC)	64	**↓ CT in 3 clusters in the bilateral SFG with large portions extending into the rMFG and rostral ACC (*p* < 0.05)** **↑ CSA clusters (*p* < 0.05):** **- left SFG/rostral ACC** **- left rMFG** **- left STG/parts of MTG** **- right SFG/ACC** **- right lOFC/rMFG**	Moderate	NA
**PET**	[62]	Relationship between cortisol plasma levels and 5-HT_1A_ receptor BP	Cross-sectional (SAD vs. HC)	30 males	**↑ cortisol plasma levels: ↓ 5-HT_1A_ BP in:** **- amygdala (*p* = 0.0067)** **- hippocampus (*p* = 0.04)**	Moderate	- Amygdala: d = 0.863 - Hippocampus: d = 0.8072
	[127]	5-HT_1A_ receptor BP	Cross-sectional (SAD vs. HC)	30 males	**↓ 5-HT_1A_ BP:** **- amygdala (*p* = 0.024)** **- insula (*p* = 0.024)** **- ACC (*p* = 0.032)**	Moderate	- Amygdala: d = 0.7892 - Insula: d = 1.292 - ACC: d = 1.131

Legend: ^1^ according to [32]; ↔ = connectivity; **↓ =** reduced; ↑ = increased; 5-HT_1A_ receptor = serotonin receptor 1A; ACC = anterior cingulate cortex; ADs = anxiety disorders; aINS = anterior insula; AUC = area under the curve; BP = binding potential; Cp = clustering coefficient of the network; CSA = cortical surface area; CT = cortical thickness; dACC = dorsal anterior cingulate cortex; DLPFC = dorsolateral prefrontal cortex; EFMT = Emotional Face Matching Task; FC = functional connectivity; fMRI = functional Magnetic Resonance Imaging; HC = healthy controls; lOFC = lateral orbito-frontal cortex; Lp = shortest path length of the network; LSAS = Liebowitz Social Anxiety Scale; MRI = Magnetic Resonance Imaging; MTG = middle temporal gyrus; NA = not applicable; PD = panic disorder; PET = Positron-Emission Tomography; PFC = prefrontal cortex; rMFC = rostral middle frontal gyrus (located within the dorsolateral prefrontal cortex); RSFC = resting state functional connectivity; SAD = social anxiety disorder; SFG = superior frontal gyrus; sMRI = structural Magnetic Resonance Imaging; STG = superior temporal gyrus; SVM = support vector machine; TVA = Temporal Voice Area. In bold statistically significant findings.

Furthermore, eight studies investigated cerebral activation during emotional face processing by fMRI. Four of these focused on amygdala activation, with moderate-QS and relatively small sample sizes. Among them, Evans and colleagues [116] explored amygdala response to angry schematic faces in 11 patients with SAD and 11 HC. The authors found enhanced responses in the right dorsal amygdala, left supramarginal gyrus and left supplementary eye field; on the other hand, activation in the right dorsal anterior cingulate cortex (dACC) appeared to be decreased in SAD patients with respect to HCs. Stein and colleagues [117] examined amygdala activation to facial expression in a small sample of subjects suffering from SAD (N = 30); results indicate increased activation in the left allocortex (amygdala, uncus and parahippocampal gyrus) with contemptuous than happy faces in SAD patients. Moreover, Yoon and colleagues [118], in another study with a small sample of subjects (N = 22), explored amygdala reactivity to emotional faces at high and low intensity; SAD patients exhibited increased left amygdala activation to high intensity emotional faces. Finally, some authors also investigated amygdala reactivity to faces at varying intensities of threat, finding enhanced left amygdala reactivity to fearful faces at moderate and high intensity, along with increased right amygdala reactivity to high-intensity emotional faces in individuals affected by SAD [119]. In addition, Prater and collaborators [120] found decreased connectivity between the right amygdala and rostral ACC at resting state, and decreased connectivity between both amygdala and the rostral ACC and dorsolateral prefrontal cortex (DLPFC) during perception of fearful faces in SAD patients. Kreifelts et al. [121] also investigated DLPFC activation during perception of laughter in 12 patients with SAD and 14 HC, demonstrating in patients an activation in the left DLPFC during reappraisal in which the negative laughter interpretation bias in SAD appeared abolished.

Remaining in the context of the limbic system, another paper [122] investigated insular response and connectivity during an emotional face-matching task in 29 patients with SAD (moderate QS). Results indicate increased activation to fearful faces in the left anterior insula (aINS) and right aINS, as well as diminished connectivity between the right aINS and dACC. Moreover, a subsequent study [123] examined the brain activation during exposure to emotional faces in 46 individuals (14 affected by SAD). Higher levels of social anxiety—as measured by higher scores on the Liebowitz Social Anxiety Scale—correlated with increased activation of the left aINS and right lateral pre-frontal cortex (PFC).

One strong-QS study [124] with a double sample (primary sample: N = 16; replication sample: N = 14) found decreased functional connectivity between the left temporal lobe and the left hippocampus for all face valances; when examined separately, the association remained statistically significant only for the angry condition.

A further study [125] examined brain activation during perception of fearful faces with a moderate QS and a small sample size (N = 26 males); the authors found that whole-brain and fear network over-activation differentiated SAD patients from HC. In addition, this research focused also on differences in regional gray matter volumes between SAD and HC, yielding significant results for the whole-brain, but not for the fear network. Structural brain characteristics were also examined in a recent study by [126] (moderate QS, N = 64), which focused on cortical thickness (CT) and cortical surface area (CSA): SAD patients showed decreased CT but increased CSA bilaterally in the dorsolateral, dorsomedial and ventromedial PFC and in the right lateral orbitofrontal cortex; CSA resulted to be increased in patients also in the left superior temporal gyrus.

Finally, one study measured 5-HT1A receptor binding potential (BP) in 12 males with SAD vs. 18 HC, using PET-imaging; results indicated lower BP in the amygdala, insula, and anterior cingulate cortex of individuals with SAD [127]. The same research group measured on the same sample the relationship between cortisol plasma levels and 5-HT1A receptor binding; the authors showed, in SAD patients, a strong negative relationship between cortisol plasma levels and 5-HT1A BP in the amygdala and hippocampus [62].

Discussion: Recently, there has been growing interest in studying both normal and pathological brain regions through a functional connectivity approach. To date, fMRI is the most extensive modality used to investigate functional connectivity in anxiety disorders, and this technique contributed to clarify the neurophysiology of aberrant emotional regulation that characterizes subjects affected by anxiety disorders [128]. Most researchers considered the amygdala and insula as main regions of interest for the analysis of the functional connectivity of patients with SAD, taking into account the important role that these brain regions play in regulation of emotions [129]. According to the limbic-cortical model, the limbic and paralimbic areas (e.g., ACC) identify personally relevant and affectively salient stimuli. The ACC, which monitors emotionally salient stimuli, connects with dorsal medial PFC and DLPFC that select, implement and supervise cognitive control strategies [130]. When functioning successfully, this network is associated with psychological resilience, flexibility and well-being. However, when not functioning perfectly, the limbic-cortical network generates acute responses that influence emotion experience, cognition and autonomic responses. Indeed, findings from RSFC studies [113,115] indicate an association of SAD with alterations in the connectivity between amygdala and frontal brain area as well as in the context of default mode network (DMN). Of note, connectivity was mostly altered in the frontal, occipital, temporal and parietal–(pre)motor regions [114,115], supporting the evidence of disrupted connections in different brain networks in patients affected by SAD. Particularly, alterations in the temporal voice–face integration area were identified as a component of biased social perception of subjects suffering from this condition [113,121,124]. In addition, SAD patients showed an excessive activation of amygdala when exposed to emotional paradigms [116,117,118,119,125]. On the other hand, less amygdala-ACC connectivity was found in patients at rest, while less connectivity between amygdala and DLPFC was observed during social threat perception (phasic abnormality) [120]. Similarly, over-activation of insula was observed in SAD, concomitantly with an aberrant connectivity of this region with prefrontal cortex during exposure to angry [123] and fear faces [122]. With regard to structural brain abnormalities, patients with SAD (compared to HC) show decreased CT and increased CSA in different areas of PFC belonging to the fear network as well as in orbitofrontal cortex and left superior temporal gyrus [126].

PET studies allowed investigating the relation between inflammation-related processes and receptor distribution in the brain. Some authors reported an inverse relationship between cortisol levels and the 5-HT1A receptors, in particular in brain regions with higher levels of steroid receptors (amygdala, hippocampus, insula and ACC) [62]. Despite the effects of HPA activation on receptor distribution in the brain, it should be better clarified what comes first: if the symptoms play a major role in neurotransmitter abnormalities, or vice versa, if inflammation and HPA axis dysregulation are the main mediators of receptor redistribution [127].

In summary, SAD is associated with different alterations in neural networks implicated in perception and emotional regulation. Despite extensive neuroimaging studies on SAD, further research is needed to corroborate the present findings that cannot be generalized in light of the small sample sizes of the studies. Nevertheless, machine-learning approaches based both on structural and functional brain imaging data could be useful to better discriminate between SAD patients and controls [124,125].

### 3.8. Neuropsychogical Markers

Three studies investigated gaze avoidance [129,130,131], reporting increased gaze avoidance in SAD patients than HC. In particular, Horley and colleagues [129] used the visual scanpath to investigate the interpersonal processing of facial stimuli in subjects affected by generalized social phobia. They demonstrated that SAD patients presented a “hyperscanning” strategy to process face expressions, with a global lack of fixations and an increased raw scanpath length, especially for neutral or sad faces. Moreover, the paper by [130] described a significant correlation between the degree of gaze avoidance and the severity of the disorder, and Weeks and colleagues [131] found that SAD patients presented a greater global gaze avoidance in comparison to non-socially anxious controls, thus supporting the hypothesis that gaze avoidance may be considered a behavioral marker of SAD.

**Table 7 ijms-24-00835-t007:** Summary of findings regarding neuropsychological biomarkers of SAD.

REFERENCE	BIOMARKER UNDER STUDY	STUDY DESIGN	SAMPLE SIZE	FINDINGS	QUALITY SCORE ^1^	COHEN’S d
[121]	Laughter perception and interpretation	Cross-sectional (SAD vs. HC)	26	**↑ negative perception of laughter (*p* = 0.005)**	Moderate	d = 1.219
[129]	Eye movement parameters (visual scanpath)	Cross-sectional (SAD vs. HC)	30	**“Hyperscanning” strategy for SAD:** **↓ n. of fixation to neutral and sad faces (*p* < 0.01)** **↓ total fixation duration (*p* < 0.01)** **↓ scanpath length for neutral faces (*p* < 0.01)** **↑ raw scanpath length for neutral and sad faces (*p* < 0.05)**	Moderate	- Total n. of fixations: d = 1.102 - Total fixation duration: d = 1.231 - Scanpath length, neutral faces: d = 0.978 - Scanpath length, sad faces: d = 0.846
[130]	Gaze avoidance	Cross-sectional (SAD vs. HC)	50	**↑ gaze avoidance:** **↓ fixations (*p* = 0.04)** **↓ dwell time (*p* = 0.03)**	Strong	- Fixations: d = 3.559 - Dwell time: d = 3.561
[131]	Gaze avoidance	Cross-sectional (SAD vs. NSAC)	39	**↑ gaze avoidance:** **↓ Total time holding eye contact (*p* = 0.04)** **↓ Number of fixation on eyes (*p* = 0.02)** **↓ Fixation durations upon eyes (*p* = 0.047)**	Moderate	- Total time holding eye contact: d = 0.283 - Number of fixations on eyes: d = 0.473 - Fixation durations upon eyes: d = 0.345
[132]	FNE, SADS, TCI, platelet 5HT_2_ receptor density	Cross-sectional (SAD vs. HC)	20 SAD males	**↑ FNE (*p* < 0.0001)** **↑ SADS (*p* < 0.0001)** **↑ harm avoidance (TCI) (*p* < 0.0001)** **↓ novelty seeking (TCI) (*p* < 0.0001)** **↓ cooperativeness (TCI) (*p* < 0.0001)** **↓ self-directedness (*p* < 0.0001)** Platelet 5HT_2_, receptor density: SAD = HC (*p* > 0.05)	Weak	- FNE: d = 4.427 - SADs: d = 4.2596 - Harm avoidance: d = 4.203 - Novelty seeking: d = 2.191 - Cooperativeness: d = 1.692 - Self-directedness: d = 4.054
[133]	Negative interpretative bias (sorting cards with emotional expressions at baseline and under threat)	Cross-sectional (SAD vs. HC)	52	**↑ probability of misclassifying neutral cards as angry under threat (*p* < 0.005)**	Strong	d = 1.835

Legend: ^1^ according to [32]; **↓ =** reduced; ↑ = increased; 5HT_2_ = 5-hydroxytryptamine receptor 2; FNE = Fear of Negative Evaluation Scale; HC = Healthy Controls; NSAC = non socially anxious controls; SAD = Social Anxiety Disorder; SADS = Social Avoidance and Distress Scale; TCI = Temperament and Character Inventory. In bold statistically significant findings.

One study exploring laughter perception/interpretation and cerebral activation patterns during fMRI found increased negative perception of laughter in 12 individuals with SAD compared with controls [121] (fMRI results are reported in the neuroimaging section).

One weak-QS study examined differences in neuropsychological scales scores—namely the Fear of Negative Evaluation Scale (FNE), the Social Avoidance and Distress Scale (SADS) and the Temperament and Character Inventory (TCI). Individuals with SAD presented higher FNE and SADS scores than HC, as well as higher TCI harm avoidance versus lower novelty seeking, cooperativeness and self-directedness [132]. The same study also explored differences in the platelet 5HT2 receptor density between patients and HC with non-significant results.

Finally, Mohlman and his group [133] explored the biased perception of facial expressions in 26 SAD patients by an emotional card-sorting task administered at the baseline and under threat (i.e., deceptive competition against another participant). This strong-QS study found an increased probability of misclassifying neutral cards as angry under threat for SAD patients.

Discussion: Different models of SAD pointed out the role of emotional hyperactivity, which arises from distorted appraisals of social situations [134]. The personality profile typical of SAD patients is characterized by a tendency to respond to perceived aversive stimuli by inhibition, in light of an inability to change behavior for the achievement of personal goals, a reduced tendency to respond to novelty by exploration and uncooperativeness in the social context [132]. Different cognitive biases among individuals with SAD were reported. For instance, these subjects could be more attuned than controls to negative emotional displays, fearful and angry faces in particular. Indeed, it was reported that individuals with SAD may misinterpret neutral facial expressions as angry in conditions of elevated anxiety [133]. On the other hand, gaze avoidance seems to be a reliable marker of the sensitivity of these patients to social concerns [129,130,131] as they exhibit this behavior at the time of receiving feedback during social interactions. Of note, SAD is characterized by a negatively biased perception of social cues and deficits in emotion regulation [135]. Cognitive reappraisal (an emotion regulation technique applied during cognitive behavioral therapy) is able to suppress SAD-associated social perception bias by involvement of DLPFC [121]. DLPFC is one of the brain areas that play a major role in emotion regulation and social perception [136].

### 3.9. Others (Neuropeptides and Electrocardiographic Parameters)

A weak-QS study [137] with a small sample size (N = 11 SAD patients) examined the plasma levels of Neuropeptide PY and norepinephrine in resting conditions and under cold stress; no significant differences between SAD and HC were found. Moreover, two studies conducted by the same research group investigated oxytocin levels in SAD. The first one, with moderate QS, evaluated baseline oxytocin levels in 24 individuals with SAD vs. 22 HC and failed to report significant differences between the two groups [138]. The second, more recent study, including a greater sample size (N = 67), also reported no statistically significant differences in oxytocin levels at the baseline (although a difference was noticeable at the trend level, *p* = 0.059); however, after the “Trust Game” (a neuroeconomic game examining trust and selfish behavior), oxytocin plasma levels resulted to be significantly lower in SAD individuals compared to HC [139].

**Table 8 ijms-24-00835-t008:** Summary of findings regarding neuropeptides and electrocardiographic biomarkers of SAD.

BIOMARKERS TYPE	REFERENCE	BIOMARKER UNDER STUDY	STUDY DESIGN	SAMPLE SIZE	FINDINGS	QUALITY SCORE ^1^	COHEN’S d
**Neuropeptides**	[137]	Plasma NPY, plasma NE	Cross-sectional (SAD vs. HC vs. PD) Resting conditions and after cold stress	11 SAD patients	SAD = HC (*p* > 0.05)	Weak	/
	[138]	Plasma oxytocin	Cross-sectional (SAD vs. HC)	46	SAD = HC (*p* = 0.8)	Moderate	/
	[139]	Plasma oxytocin	Cross-sectional (SAD vs. HC) Levels at baseline and after Trust Game	67	- Baseline: SAD = HC (*p* = 0.059) **- Mean endpoint: SAD < HC** **(*p* = 0.025)** - Change score (magnitude of change in oxytocin from baseline to endpoint): SAD = HC (*p* = 1.0) **- AUC: SAD < HC (*p* = 0.011)**	Moderate	Mean endpoint: d = 1.0361 AUC: NA
**Electro-cardiographic parameters**	[140]	QT dispersion (QTd)	Cross-sectional (SAD vs. HC)	31	**SAD > HC (*p* < 0.0001)**	Strong	d = 1.616

Legend: ^1^ according to [32]; ADs = anxiety disorders; AUC = area under the curve; HC = healthy controls; NA = not applicable; NE = norepinephrine; NPY = Neuropeptide PY; PD = panic disorder; QTd = QT dispersion; SAD = social anxiety disorder. In bold statistically significant findings.

Some authors investigated electrocardiographic parameters in a small sample of individuals with SAD (N = 16). The study has a strong QS despite the small sample size. The results showed that QT dispersion (QTd) was significantly increased in SAD patients than HC [140]. Of note, QTd is defined as the maximal interlead difference in the QT interval on the surface 12-lead electrocardiogram (ECG), and it is abnormal in various cardiac diseases. Increased QTd reflects cardiac autonomic imbalance and is associated with a higher risk of sudden cardiac death. Altered QTd was also detected in patients with severe anxiety, suggesting a predisposition for these subjects to fatal heart disease [141].

Discussion: OXT is a nine amino-acid neuropeptide, which is mainly synthesized by the magnocellular neurons in the paraventricular and supraoptic nuclei of the hypothalamus. These neurons project axonally to the posterior pituitary gland, where oxytocin is stored for release in the peripheral blood circulation. In addition, magnocellular axonal neurons also project to the prefrontal cortex, hippocampus, amygdala and nucleus accumbens [142]. Finally, OXT is released dendritically to the amygdala and sensory cortices [143]. While peripheral oxytocin facilitates parturition and nursing, centrally released oxytocin has long been known to promote maternal nurturing and mother-infant bonding. Of note, recent animal studies have underlined the importance of a subpopulation of frontal interneurons expressing the oxytocin receptor in the regulation of female socio-sexual behavior in mice [144]. In addition, oxytocin receptors in the pre-limbic medial PFC appear to reduce fear-related behaviors in pair-exposed rats [145]. Moreover, OXT was also shown to enhance social cognition and behavior, by increasing eye-to-eye gaze, interpersonal trust and the ability to infer others’ emotions from subtle social cues [146]. In the human amygdala, OXT has been hypothesized to decrease the activation of circuits involved in fear processing, hence promoting trust. Furthermore, oxytocin administration was shown to improve social symptoms in patients with autism [147] and decrease amygdala hyper-responsiveness in SAD [148]. Taken together, these considerations suggest a potential role for central OXT- and its anatomical correlates—in disorders characterized by social deficits, such as autism, or impaired social functioning, such as SAD. Several studies included in the present systematic review found alterations in the aforementioned cerebral areas, which have been implicated in regulating social behavior through oxytocin mediation. These abnormalities include increased amygdala reactivity to emotional faces [116,117,119,125], as well as reduced connectivity between the amygdala and PFC [120] in patients with SAD. Of note, impaired connectivity between these two areas might be regarded as an expression of deficient prefrontal control in the regulation of emotional stimuli. Moreover, in one study, increased dorsolateral PFC activation was implicated in cognitive reappraisal, which abolished negative laughter perception in SAD patients [121]. Reduced positive connections within the frontal lobe were also found in patients with social phobia, compared to HC [114]. Finally, during the processing of angry faces, functional connectivity between the left temporal pole and the left hippocampus was diminished in social phobics [124]. Therefore, OXT levels or OXT functioning may be altered in populations with persistent social deficits such as subjects affected by SAD; however, the topic requires further study also in light of contradicting results [138].

To date, only one study explored the role of QTd as a biomarker of autonomic imbalance associated with anxiety disorders [140]. The authors found that QTd is significantly higher in individuals with SAD compared to HC and it is also correlated with severity of symptoms. These results support the observation that anxiety disorders are associated with arrhythmias [149] and with an increased risk of sudden cardiac death [150].

## 4. Conclusions

All the interpretations and the generalizability of the results from this review should be evaluated cautiously for different limitations. First of all, most studies were based on a relatively small sample size with some of these including 40 or less participants, particularly among neuroimaging studies [62,65,109,113,114,116,117,118,119,120,121,123,124,125,127]. Second, only very few findings [63,87,94] are results from longitudinal studies. Therefore, most conclusions are limited by the single time-point, cross-sectional nature of the studies, and further research should elucidate whether some biomarkers are precursors or the effects of SAD or whether this relationship is bidirectional [80]. In this framework, large longitudinal studies recruiting high-risk individuals for SAD are needed [126]. Third, the exploratory nature of some studies, the different techniques, methods and rating scales to assess the severity of symptoms contributed to high heterogeneity between studies. Fourth, SAD is often unrecognized and diagnosed as another affective disorder [151], making it difficult to collect reliable samples of patients. Finally, different studies did not take into account confounding factors such as environmental variables [152], psychiatric and medical comorbidities [153] and medications [154]. Nevertheless, most studies were rated as of moderate quality [32].

It is also relevant to emphasize that, despite recent discoveries of biomarkers in anxiety and related disorders, different limitations have been identified. Indeed, most biomarkers to date—other than neuroimaging-based biomarkers—are assessed in the periphery (i.e., serum, plasma, saliva and urine). However, the association between central and peripheral concentrations or regulation of these parameters is extremely complex and they could be regulated or be produced from different ways. Another important limitation of peripherally biomarkers that are applied as proxies for central dynamics of the same biomarker is that the disease process itself might result in dissociation of normal physiologic relationships [155]. Finally, a significant limitation is the systematic variability in measurements, particularly with regard to neuroimaging-based biomarkers.

In summary, there were contradictions and discrepancies in findings of this systematic review, which is most likely due to the high heterogeneity among studies, small sample sizes, the lack of adjustment for known confounding factors and the relatively small amount of research done in SAD. However, our findings seem to suggest the contribution of neuroimaging studies to further the understanding of the pathological mechanisms underlying SAD. Globally, the different neuroimaging techniques applied in the studies analyzed have led to consistent or complementary results, particularly in the context of the hyperactivation of limbic structures during the processing of facial expressions of emotion and during exposition situations for the study of SAD. However, structural differences in limbic regions are not commonly used in diagnostics for economic and practical reasons [156]. Therefore, to allow the use of biomarkers and their dissemination, we would need easier and economically advantageous biomarkers. Nevertheless, the potential use of other biomarkers presented in this review is currently a suggestion that requires in depth-research. Additionally, due to the overlapping pathophysiological symptoms of mental disorders, it cannot be entirely ruled out that some biomarkers may be common to different psychiatric disorders, which may lead to an interpretation bias, therefore to the lack of highly specific biomarkers [157]. Finally, it is highly unlikely that a single biomarker for SAD can be established. However, although diagnosis of SAD is still widely based on clinical symptoms, biomarkers can be a valuable tool in order to identify individuals with such disorder from adolescence, thus improving treatment options and predicting treatment responses.

The following recommendations would highlight the significance of this systematic review and its usefulness to future topics.

1. The selection of a sample size is a determinant of experimental quality. Indeed, it is difficult to generalize the results of some studies and to draw a definitive conclusion because of the small sample size.

2. Some biomarkers discussed in this review should be evaluated in combinations to validate the main findings and to devise effective diagnostic methods. Particularly, concordance between central and peripheral compartments should be considered and characterized [158].

3. Since multimodal neuroimaging techniques are largely used in neuroscience to better visualize neural network activation, an integration of different measures can lead to more accurate data on the dynamic processes of the brain.

4. Although no putative biomarker seems to be sufficient and specific as a diagnostic tool, machine learning approaches could be useful to confirm the reliability of biological markers of SAD [159].

In conclusion, despite extensive research on SAD, further studies are needed to validate the current results and further evaluate the predictive social anxiety biomarkers compared to other.

## Figures and Tables

**Figure 1 ijms-24-00835-f001:**
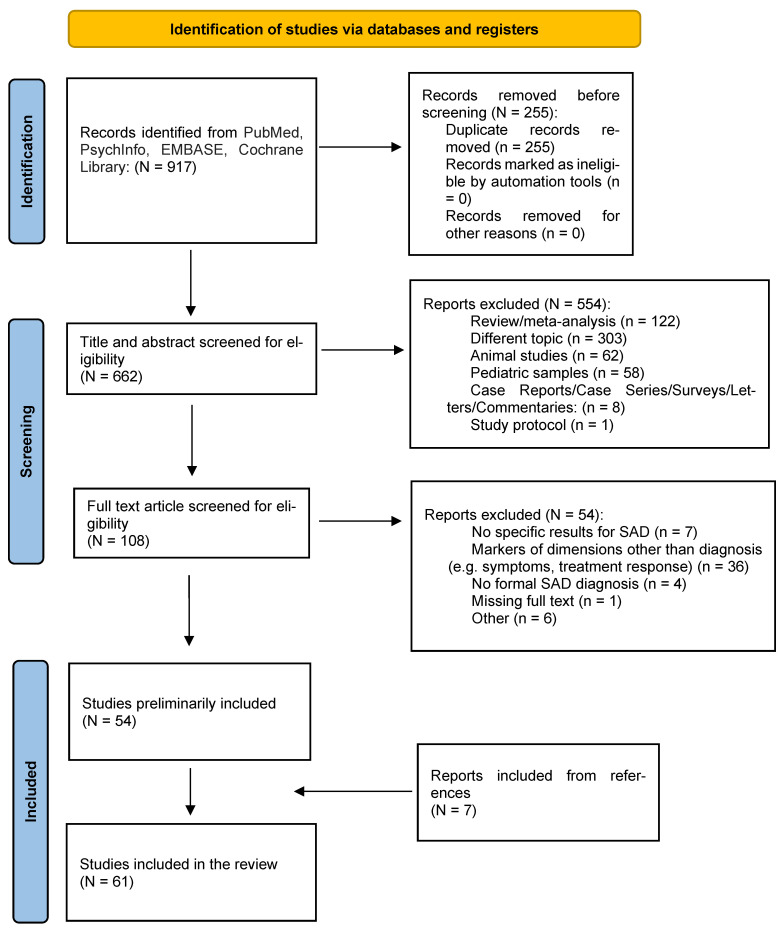
PRISMA flow diagram for systematic review.

## Data Availability

Not applicable.

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
