# Peer review of "Candidate Biological Markers for Social Anxiety Disorder: A Systematic Review"

_ijms, 2023, doi:10.3390/ijms24010835_

Round 1

Reviewer 1 Report

This work presents a review of previous works focused on biomarkers of SAD. Their methodology is valid, though some aspects need to be clarified. The introduction section needs some work, as – in my opinion – it is not complete: it is not informative enough on what are biomarkers, how they can be categorized, what previous evidence exist (on other disorders) that may sustain why exploring this in SAD also etc. There should be one discussion section, that works as an integrated whole. I provide some indications that further specify these issues.

Introduction

-        SAD has one of the lowest rates of proactively looking for help – which is possibly what the authors are refereeing to in lines 50-51. Though the importance of distinguishing biomarkers for SAD is not questioned, I still think that these biomarkers will go unnoticed if individuals don’t go looking for help and so I am not sure how recognizing these biomarkers can help early and proper treatment.

-        For the less informed reader, it might be useful if the authors provide for some framework on the biomarkers they present. For example, what is the “usual” role for oxytocin in the human functioning and, in turn, how it might be associated/ explain social anxiety. This is done in lines 211-219 as discussion, but it should be in the introduction section.

-        The introduction section should be more informative and organized into the same categories of factors the authors then use to explore their results.

Methods

-        How many individuals were involved in the selection of manuscripts? Were there disagreements? How were they solved?

-        The information that references were screened should be made clearer. Were the references taken from selected manuscripts? Were they subjected to the same inclusion and exclusion criteria? At what timing (i.e., before data was extracted from manuscripts?

-        Figure 1 is not complete so I cannot grasp its information.

Results

-        About genetics, the authors claim that previous studies used small sample sizes, but they reported one study using 321 SAD participants. So what would be a good sample size clinical sample?

-        Why do the authors present – as results – previous works with animals (lines 307-311)?

-        Lines 311-313: results were only significant for females but not for males? This is the first time results are presented based on sex.

-        The authors frequently refer to different methodologies sustaining diverse findings. It might be useful then to signal which methodologies are associated with each findings.

-        Lines 589-591 – was the sample all SAD?

-        Lines 592-613 – it is not clear what findings were specific/ diverse when comparing SAD with HC.

-        In section 3.8 – particularly its discussion – the authors are referring to areas more associated with the psychological factors associated with SAD (e.g., self-focused attention, interpretation bias). I would suggest that the authors stay closer to the papers they reviewed; instead, if going into those concepts, the authors should look deeper into the psychological models for explaining adult SAD (e.g., the cognitive model for SAD by Clark and Wells).

-         

Discussion

-        I don’t think it is justifiable to have a discussion for each section of the results. A complete and integrated discussion is called for. Moreover, these specific discussions include information that – in my opinion – should be in the introduction section of the manuscript, in as much as it introduces why some aspects might be relevant to SAD.

-        Please discuss the applicability of these biomarkers to individuals under 18 years old. This seems important given that adolescence is the principal age group for the onset of SAD.

-        Please discuss how current findings may be referring specifically to SAD and not to anxiety disorders as a whole.

-        Please provide a section with the limitations of this work (and not only of the works that were reviewed).

-        The conclusion should not be a list of limitations from the works that were reviewed. IT should highlight the main conclusions (e.g., which areas were more promising? What methodologies should be applicable, even if they have not been applied so far to SAD?)

Author Response

First of all we would like to thank the First Reviewer for the interest in our manuscript and the useful suggestions to improve the paper. We substantially revised the introduction, following your indications.

Introduction

- SAD has one of the lowest rates of proactively looking for help – which is possibly what the authors are referring to in lines 50-51. Though the importance of distinguishing biomarkers for SAD is not questioned, I still think that these biomarkers will go unnoticed if individuals don’t go looking for help and so I am not sure how recognizing these biomarkers can help early and proper treatment.

Thanks for your observation. We re-formulated the paragraph you refer to with the objective to make clearer the utility of biomarkers for SAD. Specifically biomarkers for SAD may be useful to identify subjects affected by SAD that look for help for comorbid conditions, or alternatively to facilitate differential diagnosis with other affective disorders that can share clinical symptoms with SAD. In addition, biological factors can modulate response to treatments.

- For the less informed reader, it might be useful if the authors provide for some framework on the biomarkers they present. For example, what is the “usual” role for oxytocin in the human functioning and, in turn, how it might be associated/ explain social anxiety. This is done in lines 211-219 as discussion, but it should be in the introduction section.

We added more information about the biomarkers we present (for oxytocin and the other biological factors with proper references).

- The introduction section should be more informative and organized into the same categories of factors the authors then use to explore their results.

We re-organized the introduction according to your suggestion.

Methods

- How many individuals were involved in the selection of manuscripts? Were there disagreements? How were they solved?

Thank you for the question. As reported in the Methods section, two authors checked and extracted data from articles. Disagreements were solved with the intervention of a third author who supervised the complete research activity. We reported this last information in the Methods section.

- The information that references were screened should be made clearer. Were the references taken from selected manuscripts? Were they subjected to the same inclusion and exclusion criteria? At what timing (i.e., before data was extracted from manuscripts?

Thank you for the question. We better specified these aspects in the methods.

- Figure 1 is not complete so I cannot grasp its information.

Thank you for the notice. We completed the Figure as you suggested.

Results

- About genetics, the authors claim that previous studies used small sample sizes, but they reported one study using 321 SAD participants. So what would be a good sample size clinical sample?

Thank you for the notice. We removed the sentence regarding the small sample sizes.

- Why do the authors present – as results – previous works with animals (lines 307-311)?

Thank you for the question. We reported previous work with animals in order to provide information regarding the potential anxiolytic role of steroids, based on preclinical findings. Of course, these are not part of the results. We moved the sentences at the end of paragraph on sexual steroids, but, if it remains confusing according to reviewers /editors’ opinion, we may completely remove it later.

- Lines 311-313: results were only significant for females but not for males? This is the first time results are presented based on sex.

Yes, this is a single case of results presented by sex. It is plausible that authors presented the results according to gender because the paper deals with sexual hormones, which may differ between sexes.

- The authors frequently refer to different methodologies sustaining diverse findings. It might be useful then to signal which methodologies are associated with each finding.

Thank you for the notice. We reported methodologies associated with findings into tables, with the attempt to provide a clearer combination of the aspects.

- Lines 589-591 – was the sample all SAD?

Yes, all recruited patients were affected by SAD.

- Lines 592-613 – it is not clear what findings were specific/ diverse when comparing SAD with HC.

Thank you for the observation. We reported in the manuscript the specificity (SAD/HC) of the findings.

- In section 3.8 – particularly its discussion – the authors are referring to areas more associated with the psychological factors associated with SAD (e.g., self-focused attention, interpretation bias). I would suggest that the authors stay closer to the papers they reviewed; instead, if going into those concepts, the authors should look deeper into the psychological models for explaining adult SAD (e.g., the cognitive model for SAD by Clark and Wells).

Thank you for the suggestion. Given the length of the paper, focused on biological markers of SAD, we decided not to go deeper into the psychological models for SAD and removed sentences dealing with psychological factors associated with SAD.

Discussion

- I don’t think it is justifiable to have a discussion for each section of the results. A complete and integrated discussion is called for. Moreover, these specific discussions include information that – in my opinion – should be in the introduction section of the manuscript, in as much as it introduces why some aspects might be relevant to SAD.

The discussion has been formatted according to the indication of the Associate Editor, Prof. Jessie Li - (“the structure should follow the below guide: 1. introduction; 2. methods; 3. results and discussion; 4. conclusions”). However, according to your meaningful comment, we have moved some information from Discussion to Introduction section.

- Please discuss the applicability of these biomarkers to individuals under 18 years old. This seems important given that adolescence is the principal age group for the onset of SAD.

We are grateful to the Reviewer for this important suggestion. Therefore, we added a comment on this issue in the Conclusion section.

- Please discuss how current findings may be referring specifically to SAD and not to anxiety disorders as a whole.

Following the Reviewer’s comment, we added further details on how our findings might specifically predict SAD. However, no putative biomarker appears to be specific as a diagnostic tool.

- Please provide a section with the limitations of this work (and not only of the works that were reviewed).

As previously reported, according to the indication of the Associate Editor, we merge the Limitations into the Conclusion section.  Nevertheless, as you suggested, we added further details on limitations of this work.

- The conclusion should not be a list of limitations from the works that were reviewed. IT should highlight the main conclusions (e.g., which areas were more promising? What methodologies should be applicable, even if they have not been applied so far to SAD?)

We definitely agree with the Reviewer. Consistently, we expanded the Conclusion section.

Reviewer 2 Report

This is a well-written overview about current knowledge regarding biological markers of SAD.

Just one point. The authors name oxytocin as one of the main factors, but they do not mention that oxytocin is also an identified neurotransmitter/neuromediator within human brain. "Oxytocinergic" hypothalamic fibers innervate certain brain regions, which are possibly involved in SAD. Moreover. there are reports on a subpopulation of brain frontal neurons, which even express oxytocin (and most probably innervate and thereby influence neighbouring neurons). Please consider this, when discussing putative roles of ths particular neuropeptide.

Author Response

We would like to thank the reviewer for his interest and comments on our review. We re-elaborated our section on oxytocin, following your suggestions.